# Recovery of Mixtures of Construction Waste, PET and Sugarcane Bagasse for the Manufacture of Partition Blocks

**DOI:** 10.3390/ma15196836

**Published:** 2022-10-01

**Authors:** María Neftalí Rojas-Valencia, Blanca I. Rivas-Torres, Denise Y. Fernández-Rojas, José M. Gómez-Soberón

**Affiliations:** 1Institute of Engineering, Coordination of Environmental Engineering, National Autonomous University of Mexico, Av. Universidad 3000 C.P., Mexico City 04510, Mexico; 2Department of Architecture Technology, Barcelona School of Building Construction, Polytechnic University of Catalonia, Av. Doctor Marañón 44-50, 08028 Barcelona, Spain

**Keywords:** construction waste, partition block, PET, sugarcane, bagasse, waste recovery

## Abstract

The building industry generates millions of tons of construction and demolition waste annually (12 million tons/year are generated in Mexico, of which only 4% is reused or recycled). Concomitantly, the demand for goods and services by the building industry causes significant environmental impacts. On the other hand, plastic waste is also difficult to assimilate into the environment in the short term, and its recovery is of special interest. Therefore, this research focuses on the feasibility of the manufacture of Partition Blocks (essential building element) through the combination of construction and demolition waste (CDW), polyethylene terephthalate (PET) plastic flakes, dust from tire shredding, and residue from the sugar industry (bagasse). The results of this study show that the Partition Blocks made with CDW and PET reach an average compressive strength of 115.003 kgf/cm^2^ (11.278 MPa) (suitable for structural use according to Mexican regulations); the use of lime enhances the consistency of the mixture of CDW and PET (increases its cohesion and homogeneity); and finally, these Partition Blocks have a cost comparable to the current conventional Partition Blocks made with virgin material, thus, conferring them validity as a feasible recycling option for these residues.

## 1. Introduction

For centuries, the construction industry has consumed a significant amount of natural resources such as water, and stone aggregates produced by crushing rock, soil, cement, and lime, which are extracted from mines, natural banks and rock quarries. With these materials, small and large infrastructure works are built daily around the world [1].

It is known that the building industry is a significant part of the world market, since housing is a fundamental need for human life and is essential in the social economic development of nations [2,3]. At the same time, the growing demand for goods and services in the building industry—with requirements in terms of quality, mechanical and physical properties—causes significant impacts on resources and the environment, requiring the extraction of large quantities of virgin raw materials and generating huge amounts of CDW, that reach annual figures in the order of 10 billion tons [1].

In 2017, China reported the generation of 600 million tons of CDW, while the European Union reported 500 million [4,5]. Three years later, the European Union reported 820 million tons of CDW [6,7].

In America, Canada generated 9 million tons in 2015 [8]. In the United States in 2015, 548 million tons of CDW were reported [9].

It is estimated that Mexico generates 12 million t/year of CDW, representing 30,000 t/day [10]; of these, 14,000 t/day correspond to Mexico City (NACDMX-007-RNAT-2019) [11].

According to the European Commission (2010) [12] and the FAO [13], the inadequate disposal of construction and demolition waste affects the environment and people’s safety in different ways. Due to the obstruction of natural and artificial drainage, the contamination of soils with dangerous substances, and the generation of dust and irregular human settlements, among others, there is a loss of ecosystem services. This situation leads to higher flood risks, increased respiratory diseases, and a higher likelihood of social confrontations, as well as to the loss of productivity of land dedicated to agricultural activities or land abandonment [14]. Therefore, there is an urgent need to consider CDW within the framework of measures tending to reuse and recycle waste in general [15].

In the current industrial context of the country, only 4% of the CDW is recovered (3% is recycled and 1% is reused)—the rest is taken to authorized landfills (used as filling materials), or even ends up in uncontrolled dumpsites.

There are no limits to the use of aggregates obtained by crushing rubble (mortars, bricks and concrete), since they constitute another alternative material to paving stones or facade cladding, favoring applications such as pedestrian paths in gardens for architectural landscaping and the improvement of the urban image [16]. With a view to implementing the above, Xue-Fei et al. 2021 [17] conducted experimental research on the mechanical strength and durability of chopped basalt fiber-reinforced recycled aggregate concrete.

Seco et al. (2018) [2] mention that CDW generally includes: (a) concrete from the superstructure; (b) bricks, roof and floor tiles and ceramics and blocks; (c) in smaller quantities, other materials such as glass, wood, plasterboard, asbestos, metals, plastics (mainly PET) or hazardous materials. Other authors mention similar materials [18].

According to Perera et al. (2019) [19], 60% of the generated waste consists of CDW and 5% of plastic waste (which includes 3% of PET, high-density polyethylene and polyvinyl chloride and 2% of other types of plastics) [19]. When construction waste is generated and a classification and separation of waste is not carried out, PET bottles are usually found.

On the other hand, the generation of plastic waste has caused many environmental problems due to its low biodegradability. According to several studies [20,21,22], around 300 million tons of plastic waste are produced annually around the world, 8 million tons of which end up in the oceans.

In Mexico, more than one million tons/year of used plastic ends up as waste. Even though natural biodegradable polymers are already produced as substitutes for conventional ones, the problem has not been solved, since petroleum derivatives are cheaper. It is estimated that 300 million t/year of plastics are produced in Mexico, of which only 3% is recycled [23]. Among them, we can identify PET, tires, plastic bags, packaging, household goods, furniture, pipes used in the building industry, etc., all of them being essential for the economy. These products comprise polymers, polyethylenes or polypropylenes, which are the most common polyolefin compounds—the largest group of plastics [24].

Since plastic is a problem for the environment (as well as a construction waste), it must be recycled as part of construction materials. Since PET has shown good tensile strength and a low water absorption capacity, it is recycled in the form of fibers and used as an improvement material for concrete and asphalt [25,26]. It has been seen to replace tezontle as a substitute, filling, and leveling material for irregular terrain and/or low load capacity [27,28].

On the other hand, sugarcane bagasse, the residue that remains after the production of sugar and other derivatives, is another waste that is generated in large quantities and that can be used as part of construction materials. Approximately 1 kg of sugarcane generates 25% bagasse and 0.6% bagasse ash composed of silicon and aluminum oxides, (which favors reactivity with cement.) [29]. Annually, 0.26 million tons of bagasse ash are produced [30].

The sugar industry in Mexico currently produces 20 million tons of waste per harvest, of which only two million t/year are recovered in applications such as composting or power generation [31].

Agricultural wastes such as bagasse ash are generated in large quantities in many parts of the world such as Brazil, South Africa, India, China, Cambodia, the Philippines, Indonesia, Thailand and Pakistan [32].

A more favorable alternative to handle this waste is to incorporate it into the construction industry, using bagasse ash as a pozzolanic addition to concrete and even manufacturing low-cost and environmentally friendly construction materials. According to Córdoba et al. (2018) [33], bagasse includes silicon oxide capable of reacting with other compounds and leading to an improvement of the mechanical properties and durability of the concrete. They also state that bagasse is useful for the stabilization of compacted soil blocks since it enhances their strength and durability [34] The literature reports several alternative chemical or thermal treatments for sugarcane bagasse fibers and other vegetable fibers, before their mixture with polymer matrix, cement, or plaster to improve the durability of the compounds [32].

In summary, construction materials could be manufactured from the three residues described above and, in this study, it was proposed to make Partition Blocks. Partition Blocks have been an important construction material since they have a long useful life [31]. They are one of the basic architectural elements essential for the construction of households and buildings. Their manufacturing process makes intensive use of virgin raw materials such as gravel, sand, granzón (gravel with granulometry between 3.5 and 10 mm), tepecil (inert, light and low-cost natural mineral), etc. The exploitation of mines and banks for the extraction of these raw materials entails problems such as the loss of vegetation cover, erosion, defacement of the natural landscape, damage to water resources, air pollution, noise generation, as well as the creation of lakes or pits [35,36]. Therefore, it is necessary and inescapable to prioritize the search for alternatives to replace these raw materials, as could be construction waste, PET, tire waste and bagasse.

Therefore, this research focused on assessing the integration of three types of waste, which are generated in large quantities and have a significant local incidence, such as construction and demolition waste (CDW), polyethylene terephthalate (PET), tire waste and waste from the sugarcane industry (bagasse) into a production process. Unfortunately, CDW has no particular composition since it varies according to the type of structure and/or demolition process and the construction management system [2]. However, the composition of PET and bagasse does not vary, which facilitates their use in products for the building industry, especially the manufacture of Partition Blocks, which are essential elements and represent an important volume in the construction of houses and buildings, both in Mexico and worldwide.

In view of the above, the objective of this research was to analyze the possible application of CDW, PET, tire waste and sugar mill bagasse to propose a building element with wide application in the construction industry (competitive price and comparable functional features), in order to recover all this waste.

The creation of this type of bricks arose from the idea of reusing and recycling various types of waste. The idea of the rational use of available resources instead of burying them, burning them or accumulating them in open-air dumps was also taken into account. Using recycled materials in the construction industry reduces environmental pollution, while the opposite happens when using natural raw materials.

## 2. Materials and Methods

This research was divided into four stages: field work, experimental work, normative validation of the specimens and tests conducted and economic evaluation.

### 2.1. Stage 1. Field Work

The residues used in the manufacture of the Partition Blocks were initially: CDW, PET flakes, tire dust and bagasse (from the sugar industry). The residues were obtained from industries and companies dedicated to their recovery. Details of origin and the general characteristics of the same are indicated in Table 1. The dosage of the specimen containing CDW was considered as the reference sample, and the remaining samples with percentages of other waste (PET, tires, or bagasse) with which the mixtures were completed were considered the study variables. Lime (10%) and cement (5%) were used as cementing agents at constant doses. The amount of water necessary to reach the desired consistency to allow handling and molding was also added. Finally, the samples were subjected to a 7-day curing process prior to testing.

#### Partition Block Raw Materials

The solid Partition Block comprises a mixture of the following compacted materials:(a)Natural aggregates (NA): fine and coarse (granzón—small, crushed stone, tepeci—light natural inert mineral, and sand).(b)Recycled aggregates (RA) to replace NA —representing ≥75% of the Partition Blocks components—: CDW “all in one” (crushing of waste consisting of bricks, blocks, ceramics, mortars, pavers, masonry and prefabricated elements with a granulometry ranging from ¼” to 6 mm).

Other materials considered as study variables were:(a)Polyethylene terephthalate (PET) flakes, a common type of plastic used in beverage bottles and textiles.(b)Tire dust from used tires submitted to a grinding process.(c)Bagasse—residue from the sugar industry originating from the grinding of sugarcane. The biowaste used to manufacture concrete is very diverse, among which we can find sugarcane bagasse ash, which is a biowaste from the sugar. It has been determined in laboratory studies that the elemental composition of sugarcane bagasse contains large amounts of silica, which, when reacting with free lime (after the initial reaction) in the presence of water, behaves as a binding material (as well as Portland cement), indicating that the presence of silica extends the partial substitution of cement.(d)Cement was used as a cementing material: hydraulic binder that, when mixed with aggregates (NA or RA) and water, creates a uniform, malleable and plastic mixture that sets and hardens when reacting with water (known as concrete). The usual cement for this type of Partition Blocks is Portland cement, and the one selected for this research was Portland Ordinary Cement, Resistance Class 30 of Rapid Resistance and Sulfate Resistant (Moctezuma trademark).(e)Lime was used in the same way: calcium oxide from the calcination of limestone or dolomite rocks that improves the plasticity of concrete, water retention, sand content capacity, adherence and flexibility; it also helps to avoid the efflorescence and restore microcrack damage. In this research, commercial hydrated lime supplied in bags of the “Calidra S.A.” brand was used.

### 2.2. Stage 2. Experimental Work

#### 2.2.1. Physical and Chemical Characteristics of CDW

The characterization of construction waste (6.3 mm to fines, called all in one) was carried out in accordance with Table 2.

The granulometric curve was obtained (ASTM-422-63-2020) [37]. Based on it, the particle sizes D_10_, D_30_ and D_60_ were determined and used to determine the uniformity and curvature coefficients, Cu and Cc, respectively (ASTM-D-854-02-2020) [38].

The uniformity coefficient used to evaluate the uniformity of the size of soil particles was determined. It is expressed as the ratio between *D*_60_ and *D*_10_, according to Equation (4).

Uniformity coefficient [38]
(1)$$Cu=\frac{D_{60}}{D_{10}}$$

*D*_60_ = the diameter or size below which 60% of the soil remains, by weight.

*D*_10_ = the diameter or size below which 10% of the soil remains, by weight.

Subsequently, the curvature coefficient was determined, which is defined as an indicator of the relative balance existing between the different ranges of soil particle sizes, as shown in Equation (5).

Curvature coefficient [38].
(2)$$C_{c\;\;}=\frac{{\left(D_{30}\right)}^{2}}{D_{10}\times D_{60}}\;$$

$C_{c\;}\;$ = 0.97

Cu = 15.82

From the above, it can be established that CDW can be classified as a well-graded sand with fines. They have a uniform graduation, i.e., the different particle sizes contained occupy the spaces of the holes when they are compacted, ensuring a dense and solid matrix.

The physical and chemical composition of PET, tire and bagasse was based on what is reported in the literature.

#### 2.2.2. Manufacture of Partition Blocks

Considering the information and background on the manufacture of Partition Blocks [40,41,42,43], a dosage was determined, which integrated NA, RA, cementing materials and water. The process was carried out by testing mixtures with CDW as a substitute for NA and with different percentages of additions (PET, tires and bagasse). The objective of the above was to determine the behavior of each mixture and establish the optimal ones for the manufacture of the Partition Blocks. In the process, the proportions of the materials used considered the adjustment of the granulometric characteristics of the waste, considering the principles of the manufacture of Partition Blocks [44,45,46].

The basic proportion of materials to manufacture the Partition Blocks was defined from the analyzed parameters [40,41,42,43]. The initial base dosage was the percentage of RA (CDW 80%), cement 5%, lime 10% and other waste (such as PET, tires and bagasse) 2–5%, plus 1.4 L of water per Partition Block (see Table 3).

To the previous base dosage, water was added (from 1200 to 1400 mL), depending on the humidity of the residues at the time of the manufacture of the mixtures. Water corresponds to one part of the mixture. RA, PET, tires and bagasse were incorporated according to the workability capacity of the mixtures.

A total of 12 mixtures (M1–M12) with various waste proportions were prepared. The first two contained all the residues used in this research and were used to determine the others. Table 4 presents the classification of the variables and the percentage of material used for each of them, plus 1.4 L of water per Partition Block.

In the experimental design, the statistical analysis was considered as a validation criterion, considering four factors with 12 mixtures and two repetitions. The objective was to evaluate the suitability or influence of the waste in terms of improvements of the physical and mechanical properties specified by regulations [47,48], the response variable being the compressive strength. For comparison purposes, the analysis of variance (ANOVA) was used, since it allows us to establish whether there is a significant correlation between the different samples studied or, on the contrary, if it must be assumed that their population means do not differ [31].

Once the brick mixture that showed the highest resistance value and its components were selected, its mechanical properties were evaluated through compressive strength tests, the percentage of water absorption, the initial rate of water absorption and resistance to erosion. The average drying time was also determined.

#### 2.2.3. Mixing Process

Mixing is one of the most important parts of the process, since 70% of the final quality of the product depends on it [31]. The mixing of aggregates, cementing agents and water provide as a result the molding mixture, which, with the appropriate balance of raw materials, allows for adequate workability and a product with the proper quality finish (uniform surface and material distribution). Figure 1 shows the manual mixing. The sequence of the process consisted of adding the aggregates, cement and lime, mixing them manually in a dry state, then adding water. The suitability of the mixture for molding purposes was established (five iterations of each mixture [31] were carried out); finally, the molding was performed according to the manufacturing procedure [31,49].

#### 2.2.4. Molding of Specimens

The equipment used to manufacture the Partition Blocks was artisanal machinery (Rojas-Valencia, 2022 patent number 388601) [50], in which it is possible to mold two specimens at the same time, as shown in Figure 2. Each of the two resulting specimens are prisms with the following dimensions: 6 × 12 × 26 cm.

#### 2.2.5. Setting

The setting time for the material was an average of two days at room temperature between 20 and 27 °C. Water and humidity were kept at adequate levels to avoid cracks or deformations of the material, thus, ensuring an adequate strength [31].

#### 2.2.6. Partition Blocks Manufacturing Process

The manufacturing was carried out according to the process shown in Figure 3. The stages were: (a) recovering waste and raw materials from the various industries, as well as the cementing agents; (b) weighing the materials in accordance with the design dosages for each mixture; (c) manually mixing the residues with the cementing agents until the adequate molding consistency is obtained; (d) molding the Partition Blocks using the artisanal machinery; (e) keeping the Partition Blocks in ambient humidity conditions to avoid cracks (constant monitoring); (f) weighing and measuring the Partition Blocks [49]; (g) drying; (h) performing the quality tests established by regulations (resistance and initial water absorption) [47,48].

### 2.3. Stage 3. Normative Validation of the Specimens and Tests Conducted

The summary of the applicable regulations for the Partition Blocks tests was based on the Mexican regulations, which are shown in Table 5, since this material is manufactured for Mexico. An exhaustive review of national and international regulations was made, and no standard could be found which referred to Partition Blocks manufactured with the mixture used in this research.

The mechanical properties were evaluated through compressive strength tests, the percentage of water absorption, the initial rate of water absorption and resistance to erosion. The average drying time was also determined.

#### 2.3.1. Geometry of the Partition Blocks

The sizing of the specimens was based on the NMX-C-038-ONNCCE-2004 standard [49] and was determined by using manual equipment with a calibrated scale (Calliper brand vernier) with an approximation of 0-150 mm. The value obtained is the average result of 5 measurements made in different areas of each dimension of the specimens. The dimensions indicated in Figure 4 were determined for each specimen. The minimum acceptable nominal dimensions are as follows: thickness 6 cm, width (header) 12 cm and length (stretcher) 26 cm; all of them are ±3 mm.

#### 2.3.2. Determination of the Initial Water Absorption

The sequencing to determine the initial water absorption (NMX-C-037-ONNCCE-2013) [48] is detailed in Figure 5. The formulation applied to obtain the initial water absorption coefficient was through Equation (1) [48].
(3)$$C_{b}=\frac{100M}{S\sqrt{t}}=\frac{100\left(M_{1}-M_{s}\right)}{S\sqrt{10}}$$
where:
*Cb* is the initial absorption coefficient in g/(cm^2^ × min).*M* is the mass of water absorbed by the block during the test in gr. (*M* = *M*_1_ − *M_s_*)*M*_1_ is the wet mass in g.*M_s_* is the dry mass in g.*S* is the surface of the submerged face in cm^2^.*t* is the immersion time in min (t = 10 min).


#### 2.3.3. Determination of Compressive Strength

Figure 6 shows the sample preparation sequence of the procedure for performing the compressive strength test (NMX-C-036-ONNCCE-2013) [47]. The test age was established at 28 days after the manufacture of the specimens, considering that the cementing materials need this period to reach their specification strength. Equation (2) is used for its determination [47].
(4)$$R=\frac{F}{A}$$
where:
*R*: compressive strength in MPa (Kgf/cm^2^).*F*: maximum load in N (Kgf).*A*: cross-sectional area of the specimen (cm^2^).


#### 2.3.4. Water Absorption Test in Percentage (24 h)

The determination of the water absorption in percentage per 24 h was carried out using the procedure described in Figure 7 based on the standard. It was determined using Equation (3) [48].
(5)$$A=\frac{M_{SSS}-M_{s}}{M_{s}}\times 100\%$$
where:
*A*: the volume of water absorbed referred to the apparent volume of the specimen in dm^3^/m^3^.*M_s_*: dry mass of the specimen in g.*M_sss_*: saturated and superficially dry mass in g.


The factor 100 is applied as a unit converter dm^3^/m^3^ (equivalent to liters per m^3^).

#### 2.3.5. Erodibility Test

Strength and stability are known to be lost in land-based building materials when they are in contact with water for extended periods of time. Therefore, in several studies, the spray erosion test is used to assess the durability of earthen blocks [44].

On the other hand, erodibility is established as an additional test for terrestrial construction materials, according to the New Zealand standard NZS-4298-1998 [53]. The test consists of spraying water on one of the faces of a block for one hour or until the water penetrates the sample. Every 15 min, the test is interrupted to check the depth of the erosion caused by the water in the block.

The maximum depth is measured one hour after the start of the test. When water bores a hole through the sample in less than one hour, the erosion rate is obtained by dividing the thickness of the sample by the time it takes for full penetration to occur. The erodibility index is determined according to Table 6:

### 2.4. Stage 4. Economic Evaluation

The evaluation of projects has become a priority among the economic agents that participate in any of the stages of allocation of resources to implement investment initiatives. For this reason, in this work, an economic evaluation was carried out for the manufacture of blocks at a commercial level. The State of Puebla, Mexico, was taken as a reference, since it is the entity that, with around 4500 brick kilns in its territory, has the largest number of artisanal brick kilns in the country, i.e., approximately 40% of the brickyards in Mexico [54].

## 3. Results

### 3.1. Physical and Chemical Characteristics

#### Physical and Chemical Characterization of CDW

The results of the physical and chemical characterization of the CDW used to make the mixture are shown in Table 7.

As regards the physical characteristics of CDW, sand is the major mineral in the construction materials (94%), while fines represent only 6%.

### 3.2. Granulometric Curve of Construction and Demolition Waste

Figure 8 shows the granulometric curve that was obtained (ASTM-422-63-2020) [37]. As already described in the methodology, the particle sizes *D*_10_, *D*_30_ and *D*_60_ were determined and used to determine the uniformity and curvature coefficients, Cu and Cc, respectively (ASTM-D-854-02-2020) [38].

From the above and Equation (5) ($C_{c}$ = 0.97 and Cu = 15.82), it can be established that CDW can be classified as a well-graded sand with fines. They have a uniform graduation, i.e., the different particle sizes contained occupy the spaces of the holes when they are compacted, ensuring a dense and solid matrix. Table 8 show the results of the physical and chemical characteristics of PET.

The physical properties of polyethylene terephthalate (PET) as a thermoplastic, as well as its crystallinity, allow its transformation in different ways, such as extrusion, injection, injection and blowing, thermoforming processes, among others.

Its physical properties, such as resistance to wear and corrosion, chemical and thermal resistance, waterproof characteristics, resistance to permanent efforts and wear, low moisture absorption, and high surface hardness make this material a valuable ingredient of the mixtures used to manufacture blocks and other construction materials.

PET has a chemical resistance to hydrocarbons, alcohols, fats and oils, ether, diluted acids, and acids. The benzene ring not only provides increased rigidity, but also greater chemical resistance [23,55,56].

#### The Composition of Sugarcane Bagasse in Percentage Is as Follows:

The composition of bagasse is mostly cellulose, hemicellulose, lignin and other minor constituents [57]. As regards fibers, 65% correspond to pure fibers, while 35% correspond to bagacillo. The composition of the sugarcane bagasse is shown in Table 9.

Table 9 shows data generated in a study on sugarcane bagasse. According to the research and previous information, we can say that bagasse is a low-density fibrous material with large-size particles and a high moisture content [58].

Kazmi et al. (2016) [29] determined that the manufacture of fired clay bricks that incorporate bagasse ash showed a lower unit weight, which resulted in lighter bricks and a better performance in earthquake-prone areas. Moreover, in another study by Junaid et al., (2021) [59] an improved thermal performance was observed with clay bricks incorporating biomass ash. The substitution of clay by 15% bagasse ash and RHA (rice husk) in the production of fired clay bricks reduced thermal conductivity by 31% and 29%, respectively.

The use of bagasse ash in bricks is very encouraging in terms of waste disposal. However, the use of higher concentrations (i.e., 10%) of bagasse ash in clay bricks tends to decrease the compressive strength [29].

### 3.3. Compressive Strength for the First 12 Mixtures

NMX-C-ONNCCE-404-2012 standard [51] establishes the minimum compressive strength value of 100 kgf/cm^2^ (9.807 MPa) to consider the Partition Blocks acceptable to be used as structural elements. The dosage leading to the highest compressive strength—and the only one that exceeds the normative requirement—of the different dosages studied corresponds to M10-PET 5%, with a compressive strength value of 115.8 kgf/cm^2^ (11.35 MPa). Figure 9 shows the results of the tests of the 12 mixtures.

Although the rest of the mixtures are below 100 kgf/cm^2^ (9.8 MPa), NMX-C-ONNCCE-441-2013 standard [52] establishes the minimum resistance limit of 30 kgf/cm^2^ (2.942 MPa) for Partition Blocks to be used as non-structural elements. In this case, all the mixtures studied meet this requirement—therefore, all the mixtures studied can be used in the building industry. These results are in line with previous studies [42,43,44,45,46,47].

The values indicated in Figure 9 are the average compressive strength of the Partition Blocks made from the various mixtures. The results establish the feasibility and mechanical performance of construction waste: “all in one” 6.3 mm, one quarter to fine, and residues (PET, tires and bagasse) used to replace stone aggregates within the matrix of a Partition Block. The M10-PET 5% mixture with 80% CDW “all in one” and 5% PET flakes obtains the best mechanical behavior (11.4 MPa) and exceeds the minimum compressive strength established by NMX-ONNCCE 404-2012 [51] for structural use, compared to others that only comply with NMX-C-441-2013 [52] for non-structural use (9.81 MPa). This coincides with what is reported in other research [19,20,21,55,56] and seems to be related to the suitable uniformity and cohesion achieved with this mixture, unlike what is observed with the others.

Although the use of bagasse ash as a cementitious material reduces the amount of superplasticizer required and improves the yield strength and fluidity, it is possible that the durability of the final bio-waste-based concrete products is affected, since by increasing the percentage of sugar cane bagasse, its resistance is decreased, due to the absence of free lime produced during the chemical composition [29,60,61].

In this study, bagasse concentrations were minimal, ranging from 2% to 5%. Despite this, the resistance of the Partition Block was affected by the compressive strength when using the bagasse, which confirmed what other researchers observed [29,59,60,61].

Figure 10 shows the surface appearance of a Partition Block obtained with mixture 10 (M10-PET 5%). Its adequate physical characteristics can be seen: it is homogeneous, uniform, monolithic and without cracks.

Based on the above results, mixture number 10 (M10-PET 5%) was determined as the optimum mixture for the manufacture of Partition Blocks, with extreme values of variability in compressive strength ranging from 113 to 118 kgf/cm^2^ (from 11.081 to 11.571 MPa) (always suitable as elements for structural applications), an efficient workability of the mixture during manufacturing, and with the aforementioned physical characteristics. Table 10 shows the dosage used for the best mixture.

Five specimens per test variable were made with this dosage, according to the optimal mixture M10-PET 5% (from M10-PET 5%-R1 to M10 PET 5%-R5), in addition (as evidenced) to five specimens with the reference CDW base mixture M3-CDW: (from M3- CDW-R1 to M3-CDW-R5) “all in one” (6.3 mm) without the addition of PET, as shown in Table 11.

### 3.4. Compressive Strength Test of the Partition Blocks Obtained with the M10-PET 5% Mixture

The results of the compressive strength tests on the Partition Blocks obtained with the M10-PET 5% mixture and the reference ones are shown in Figure 11. It can be seen that the blue provides the best result and those in grey do not fulfil the standard. We confirm that when PET is aggregated to the mixture, the strength is higher. The values for the five specimens exceed the minimum value established by the standard, with an average of 111.7 kgf/cm^2^ (10.954 MPa); for reference specimens (CDW only) this is 76.897 kgf/cm^2^ (7.541 MPa).

The results shown in blue are from the compressive strength tests of a mixture that incorporates 5% PET, and the results in grey are from a mixture that replaces that 5% PET with 5% CDW.

Partition Blocks comprising the mixture of CDW 80% with PET 5% exceed the minimum required resistance established in the NMX-ONNCCE 404-2012 [51] for structural parts. While Partition Blocks comprising CDW 85% do not reach the minimum resistance required for structural parts, they do comply with the standard for non-structural parts.

Statistical comparisons between specimens of CDW 85% and CDW 80% with PET 5% showed significant differences and a higher compressive strength with the addition of PET flakes, in various studies in which PET was used to manufacture Partition Blocks, blocks and bricks. According to Zhang 2013 [18], who studied the manufacture of bricks and tiles from plastic waste (PET), the incorporation of an additive led to a higher compressive strength, reaching a value of 40.78 kgf/cm^2^ (4 MPa); moreover, the addition of residues helped to decrease the absorption percentage, as shown in the present paper.

Concrete with up to 30% higher compressive strength has been manufactured with construction and demolition waste compared to concrete made with conventional aggregates [31,35].

Figure 12 shows the standard deviation and the average of the M10-PET 5% Partition Blocks. The upper limit and the lower limit represent the maximum and minimum values according to the standard deviation of the resistance of the samples. The average strength is 10,954 MPa., the standard deviation is 0.82 MPa., the lower limit is 10.134 MPa and the upper limit is 11,774 MPa.

From previous studies [40,55,56], in which Partition Blocks were manufactured from a mixture of sand, cement, lime, and PET as a coarse aggregate, it is clear that the addition of lime improves the characteristics of the mixture, preventing the disintegration of the components, while the decrease in the percentage of PET contributes to maintaining the stability and strength of the Partition Blocks. In this research, an average compressive strength value of 99.83 kgf/cm^2^ (9.790 MPa) was obtained, compared to a usual compressive strength ranging from 110 to 115 kgf/cm^2^ (10.787 to 11.27 MPa) reported by many laboratories for Partition Blocks.

This research confirms that the addition of lime increases the adherence between the materials. The percentage of PET is essential for the final resistance of the Partition Block, since high amounts of the residue do not allow a homogeneous mixture, leading to an inappropriate quality [19,20,21].

However, it must be taken into account that investigations exist indicating that lime (CaO) reacts violently with water to form an alkaline suspension, and releases a large amount of heat, which in turn promotes the reaction of the air-entraining agent to form bubbles. This process greatly affects the final porosity and pore structure of the cellular concrete, causing the high heat of hydration and early expansion of the final products [62]. In this study, these effects were not seen, possibly because the percentage of lime used was low.

Various studies have shown that PET can be used in this industry, where it has been applied as masonry, gabion fills, the partial or total replacement of slopes and as a support material in retention structures, among others [19,20,21,25,26,27,28].

Partition Blocks made from construction waste and PET contribute to avoiding the exploitation of banks of natural aggregates, which is a serious environmental and social problem [31].

### 3.5. Initial Water Absorption

Water absorption was determined for the study dosage (M10-PET-5%) and reference dosage (M3-CDW) in accordance with the standard NMX-C-037-ONNCCE-2013 [48] and with the results of the standard NMX-C-404-ONNCCE-2012 [51], maintaining the samples in dry conditions in a desiccator manufactured specifically for this research, at a constant temperature of 100 °C. Figure 13 shows the equipment and specimens in the testing process.

Figure 14 shows the specimen being submitted to the water absorption test and the reference samples; the acceptance criteria of the NMX 037-ONNCE 2013 standard [48] indicates a maximum of 5 (g/min) of water absorption.

Figure 15 shows that the mixture M10-PET 5% with PET becomes less impermeable 0.6% (g/min), while the one containing only construction waste is more permeable than what is established in the Mexican standard NMX- 441-ONNCCE-2013 [52]. Therefore, PET can help reduce permeability problems, which is in accordance with what has been said previously.

### 3.6. 24-h Water Absorption Test

In accordance with the necessary tests, the water absorption test was carried out at 24 h on the Partition Blocks made with mixture 10 (M10-PET-5%). Figure 16 shows that all the results are below the value established in the norm, the percentage of water absorption of the Partition Blocks being thus acceptable. The M3-RCW mix shows results of 8.1 g/min. This means that it is highly permeable and does not meet the water absorption percentage stipulated in the standard. Mixture 10 (M10-PET-5%) shows water absorption results of 4.4 (g/min), below the limit of 5 g/min established by the standard, and thus, the water absorption percentage of the Partition Blocks is acceptable.

In all the tests, the results of the total maximum water absorption percentage remained below what is established by the Mexican standards. It should be noted that there is no specific regulation for this new mixture of PET and CDW.

The tests carried out for the percentage of water absorption at 24 h show an average value of 20.116%, which means that it complies with the percentage of absorption (%) at 24 h (27%) set forth in Mexican standard NMX-C-037-ONNCCE- 2013 [48]

This is very favorable for these construction materials. In addition to the above, these results coincide with what has been reported in other research [23,55,56,57,58,59].

### 3.7. Erodibility Test

Finally, in the erosion test, the Partition Blocks of the study exhibited a resistance to water penetration more than twice as high, compared to the blocks that are for sale in authorized establishments. This remarkable result was due to the tire dust and PET content.

### 3.8. Economic Evaluation

Regarding the economic analysis, the manufacturing cost of Partition Blocks with PET is similar to that of using natural raw materials, but long-term benefits can be observed through the reduction in this waste in landfills and the absence of the need to extract aggregates for the manufacture of Partition Blocks.

According to the study carried out to determine the economic feasibility of manufacturing blocks with CDW and PET at commercial scale, it was determined that with an estimated production of 5000 pieces per month, a period of 5 years would be necessary for the block maker to recoup their investment.

The cost of a block with CDW would be USD 0.41, including operating and investment expenses, and at a commercial sale price of USD 0.59, profits of USD 933.98 would be obtained with a sale of 5000 Partition Blocks per month.

It was determined that an initial investment of USD 16,418.70 was needed for the acquisition of the mechanical block molder, the property and a vehicle for distribution purposes, office equipment, communication equipment, machinery and the equipment of the block company, tools, facilities and civil works.

The following materials are necessary for manufacturing 1000 Partition Blocks: 6.4 tons of CDW all in one, 400 kg of PET flake, 400 kg of cement and 800 kg of lime.

According to the study, the cost per piece of a Partition Block, with the substitution of natural stone aggregates for waste, is viable and competitive on the local market. According to the National Institute of Statistics and Climate Change (INECC), the age of more than 80% of the companies dedicated to the production of bricks and Partition Blocks does not exceed 30 years, and 20% of them are 5 years old or younger. Among those who market, 66% do so directly to the consumer and 34% resort to intermediaries.

In contrast, artisanal brickmakers do not have sufficient bargaining power to influence the prices paid by intermediaries and do not have access to an increase in profit margins through direct marketing. The sale of the blocks has a higher profit margin for the manufacturer who markets them directly. The combination of manufacturing and commercialization activities of bricks and Partition Blocks in the same site allows the use of freight transport to cover larger distribution areas of the product.

The main problems faced by 51% of the Partition Block and brick producers are the supply of raw materials, the maintenance of the machinery used in their production, the quality of the inputs and their costs. In this way, the substitution of natural aggregates for construction waste contributes to the solution of the main problem, which is the supply of raw materials.

In addition to the environmental advantages generated by the manufacture of Partitions Blocks made with construction waste and PET, it is worth noting that the use of these materials can deliver economic rewards to the local construction sector, since they are competitive on the market [23,55,56].

For a wide production and application of Partition Blocks manufactured from waste materials, more research and development are needed, not only in the technical, economic and environmental areas, but also in standardization, government policy and public education related to recycling, waste and sustainable development [2,8,16,17,18,19,30,31,32,33,34].

Sustainable Partition Blocks can be used in the construction of partition walls and exterior walls, since the tests have shown that they can also be used in load-bearing walls.

Table 12 shows the comparison between Partition Blocks made from natural aggregates and Partition Blocks made from recycled materials.

Due to the importance of finding new alternatives to supply the construction industry with products and materials manufactured from CDW that are generated in significant quantities and currently only somewhat reused or recycled, the manufacture of the Partition Blocks proposed in this paper is viable.

## 4. Conclusions

The evaluation of the quality of the Partition Blocks according to the Mexican standards generated satisfactory results that met the technical parameters. The use of CDW (more than 80% of the mixture) as a substitute for stone aggregates in common Partition Blocks with the addition of PET stands out significantly, since this contributed to raising parameters such as compressive strength above (11.77 MPa) and to decrease absorption (5 g/min).

According to the results obtained in the present study, using the M10-PET 5% dosage, CDWs are an alternative to replace 80% of natural aggregates. Moreover, because of its physical properties, the addition of PET contributes to improve the compressive strength of the specimens. Lime is favorable in the mixture of CDW with PET because it increases the cohesion of the mixture and improves its homogeneity.

The specimens obtained in the present research reached acceptable compressive strengths for structural (those containing PET M10–M12) and non-structural use (the rest of the mixtures M1–M9).

The results of the compressive strength limit of Partition Blocks using tire dusts provided resistance below the standard NMX-C-ONNCE 404, the best dose being M5-LLAN 2.5% (8.3 Mpa). While the results using sugarcane bagasse gave the lowest results, in all cases, the Partition Blocks, according to the regulations, could be used for non-structural use.

With respect to the maximum initial water absorption for the Partition Blocks test, only the mixture M10-PET 5% with PET becomes less impermeable 0.6% (g/min), while the rest of the mixtures were more permeable than what is established in the Mexican standard NMX-441-ONNCCE-2013.

The tests carried out for the percentage of water absorption at 24 h show an average value of 20.116%, which means that it complies with the percentage of absorption (%) at 24 h (27%) set forth in Mexican standard.

In the erosion test, the Partition Blocks (mixture M10-PET 5% with PET) of the study exhibited a resistance to water penetration more than twice as high, compared to the blocks that are for sale in authorized establishments. This remarkable result was due to the tire dust and PET content. In the partitions with sugar cane bagasse, the erosion was faster.

For manufacturing 1000 Partition Blocks, the following are necessary: 6.4 tons of CDW all in one, 400 kg of PET flake, 400 kg of cement and 800 kg of lime.

The cost of a block with CDW would be USD 0.41, including operating and investment expenses, and at a commercial sale price of USD 0.59, profits of USD 933.98 would be obtained with a sale of 5000 Partition Blocks per month.

According to the study, the cost per piece of a Partition Block, with the substitution of natural stone aggregates for waste, is viable and competitive on the local market.

## Figures and Tables

**Figure 1 materials-15-06836-f001:**
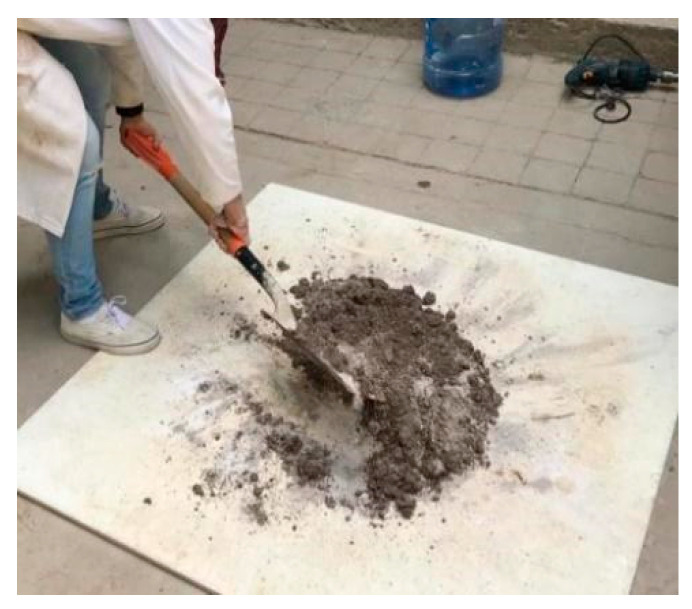
Mixing waste and cementing agents for Partition Block manufacturing.

**Figure 2 materials-15-06836-f002:**
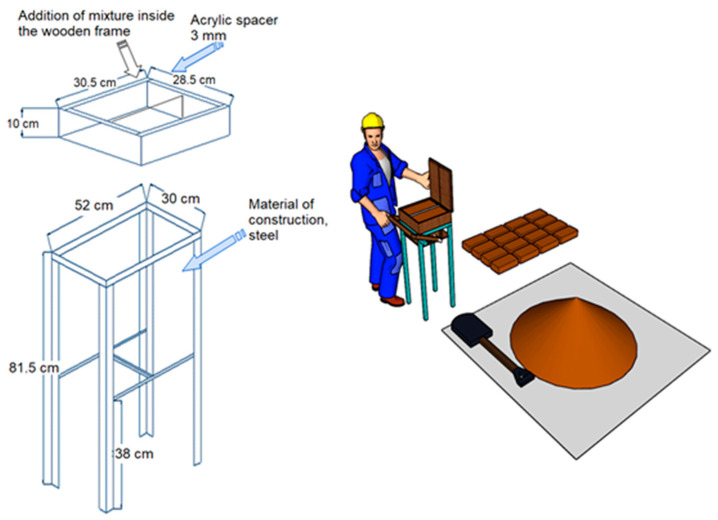
Brick-molding machine.

**Figure 3 materials-15-06836-f003:**
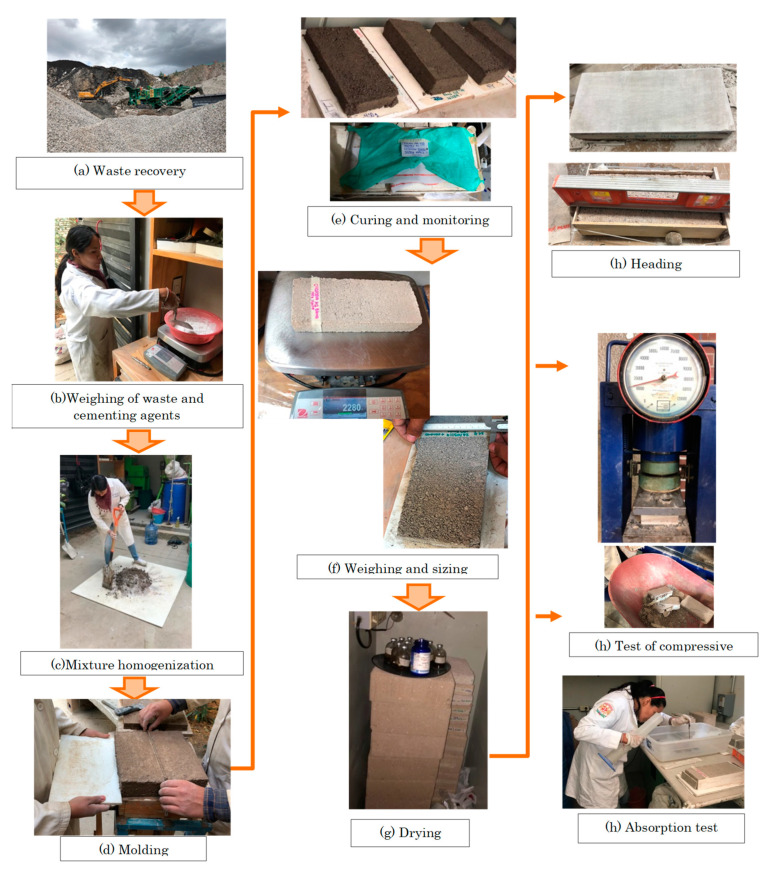
Specimen manufacturing process.

**Figure 4 materials-15-06836-f004:**
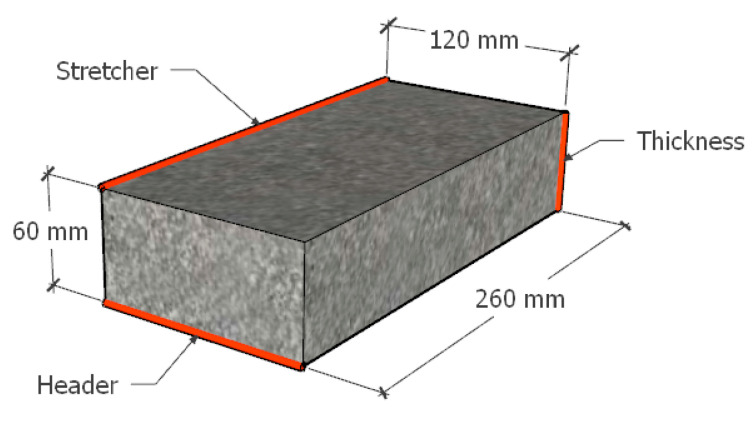
Dimensions and parts that comprise the manufactured specimens: the red lines describe geometric denomination by normative, and the black lines describe real Partition Block dimensions.

**Figure 5 materials-15-06836-f005:**
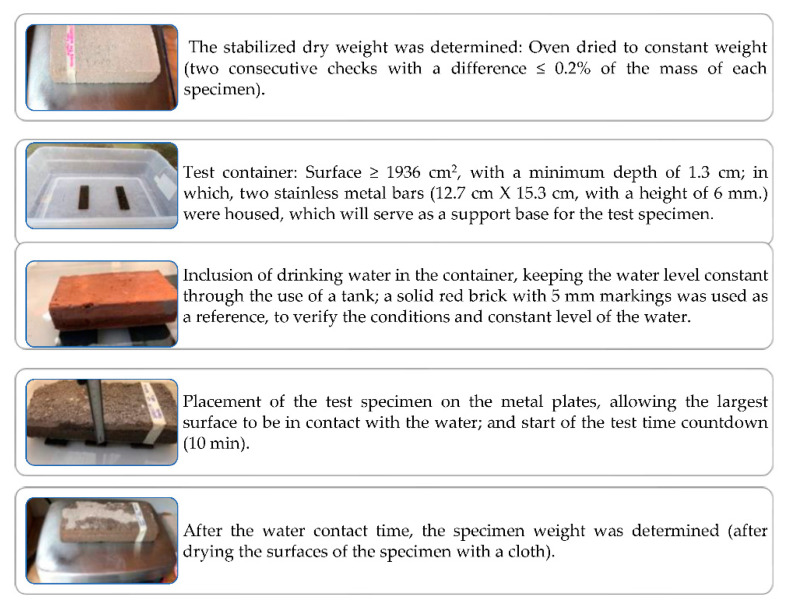
Initial water absorption test sequencing.

**Figure 6 materials-15-06836-f006:**
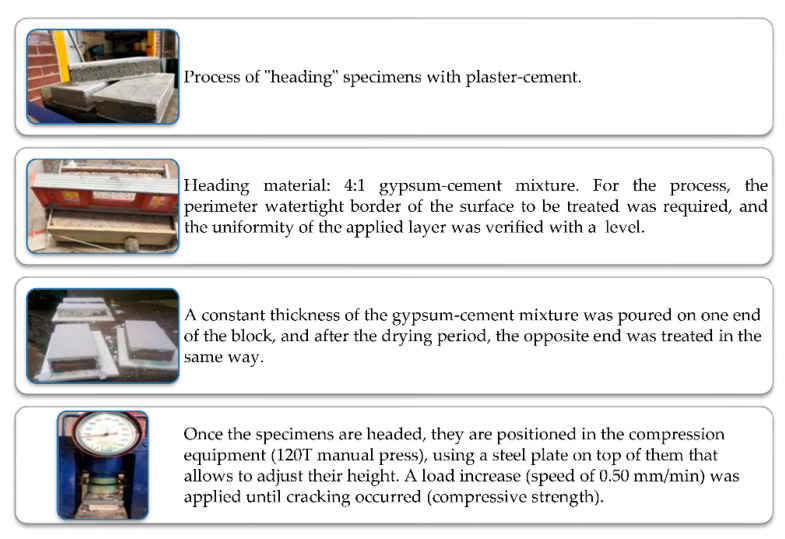
Sequence of preparation of specimens for compression test.

**Figure 7 materials-15-06836-f007:**
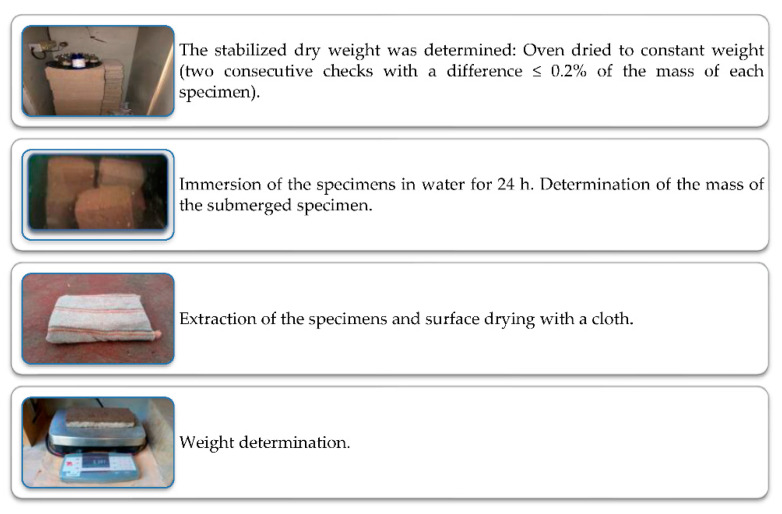
Determination sequence of the water absorption test at 24 h.

**Figure 8 materials-15-06836-f008:**
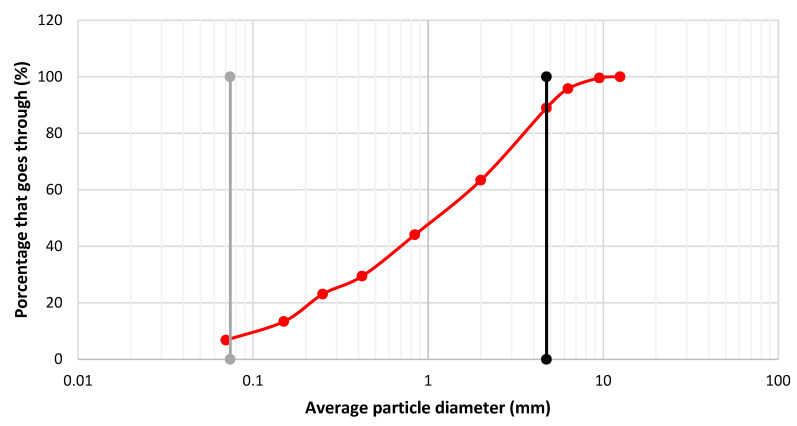
Granulometric curve of construction waste, from one quarter to fines.

**Figure 9 materials-15-06836-f009:**
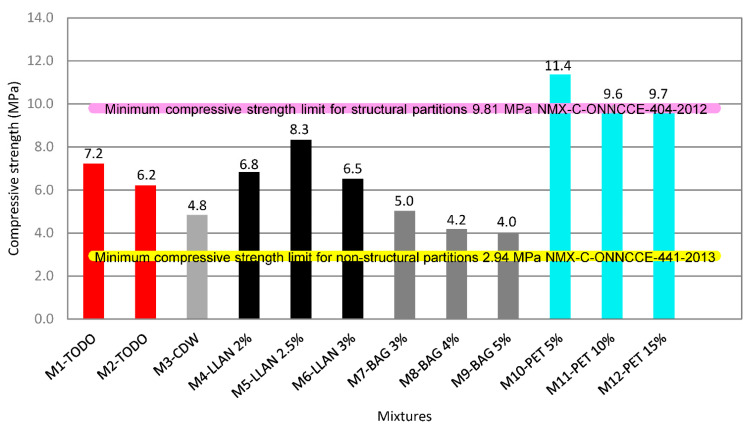
Results of compressive strength of the different Partition Blocks studied.

**Figure 10 materials-15-06836-f010:**
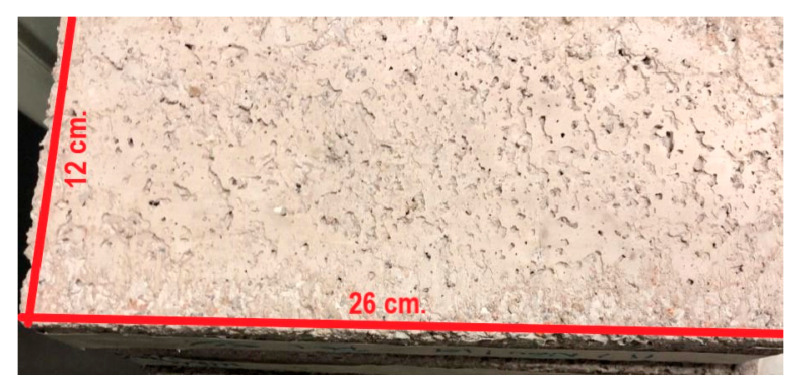
Partition Block obtained with mixture 10 (M10-PET 5%).

**Figure 11 materials-15-06836-f011:**
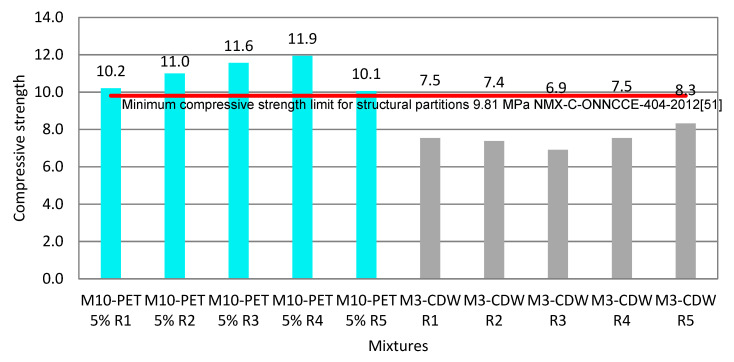
Compressive strength of PET and CDW Partition Blocks.

**Figure 12 materials-15-06836-f012:**
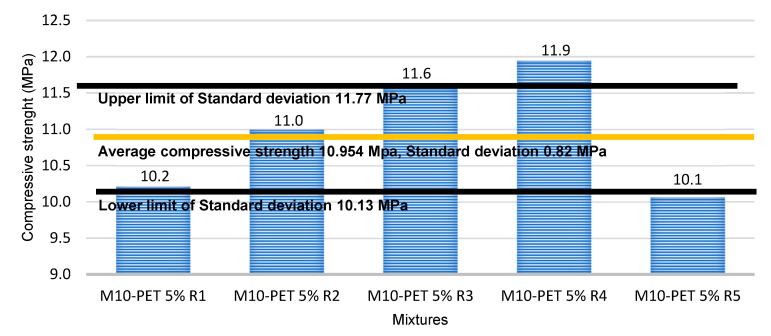
Standard deviation and average compressive strength of the Partition Blocks made with CDW-PET 5%.

**Figure 13 materials-15-06836-f013:**
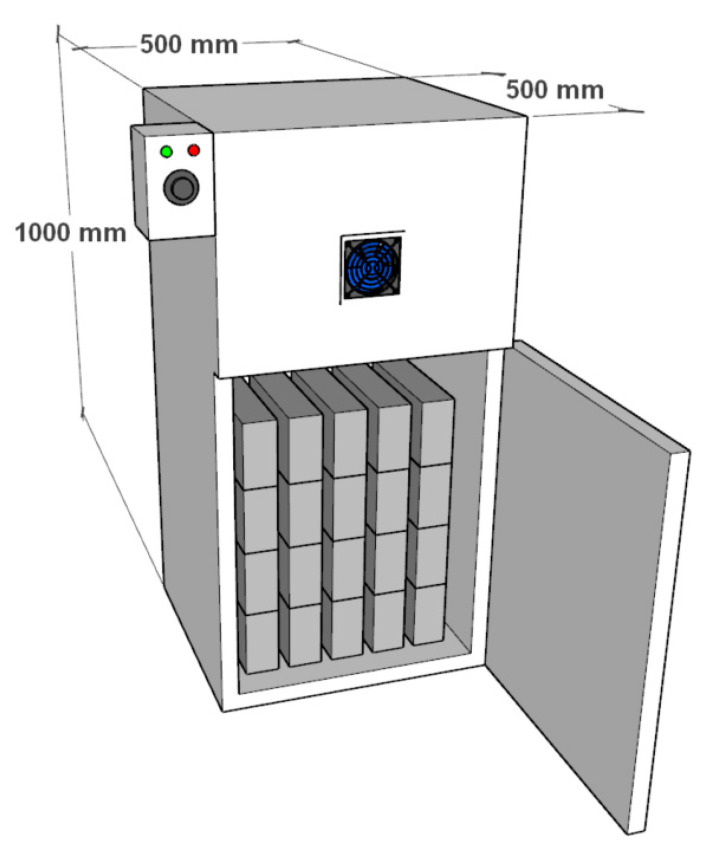
Partition Block drying equipment for the water absorption test.

**Figure 14 materials-15-06836-f014:**
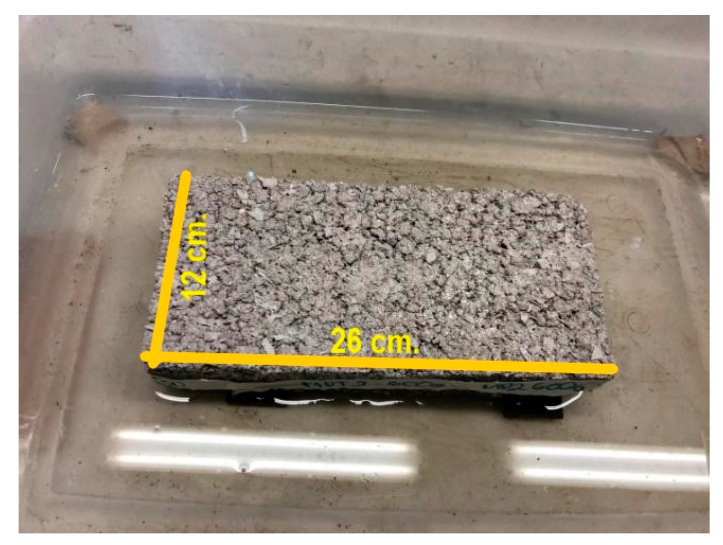
Specimen being submitted to the water absorption test.

**Figure 15 materials-15-06836-f015:**
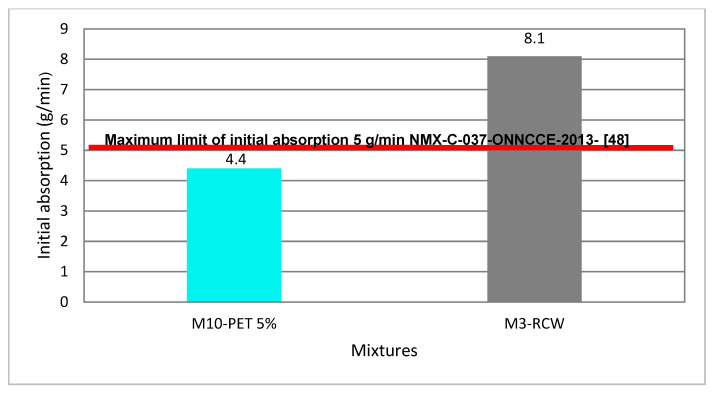
Maximum initial water absorption for Partition Blocks made with CDW-PET 5% and CDW.

**Figure 16 materials-15-06836-f016:**
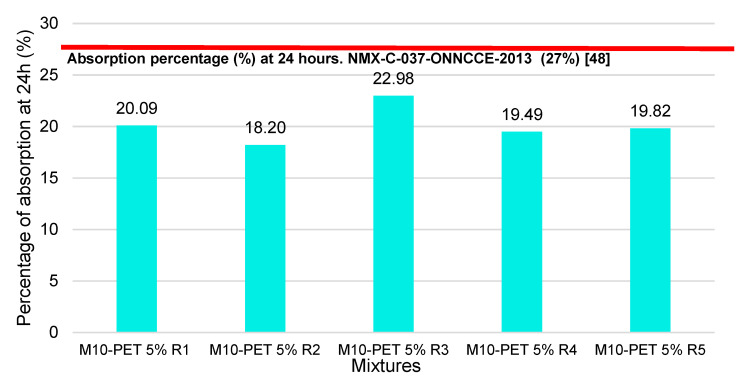
Water absorption percentage at 24 h for Partition Blocks made with CDW-PET 5%.

**Table 1 materials-15-06836-t001:** Recovered waste.

Waste	Characteristics	Waste Origin	Picture
Construction waste “¼ all in one”. (6 mm)	Mixture of demolition waste (buildings, excavations, roads, urban developments, pathways, etc.)	Obtained from the Company Concretos Reciclados S.A. de C.V., Mexico	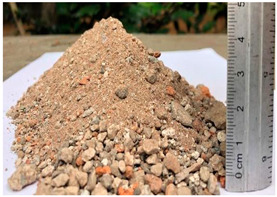
PET flakes	Waste from the mechanical shredding of post- consumption PET bottles.	Obtained from the company Hojuelas de PET, México	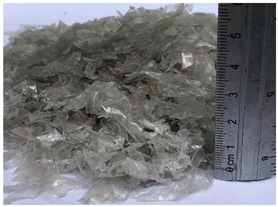
Tire dust	Rubber dust or tire dust mesh 20; the waste derives from the mechanical shredding of used tires (NFU) purity: 99%.	Obtained from the company Genbruger products from recycled tires, Mexico.	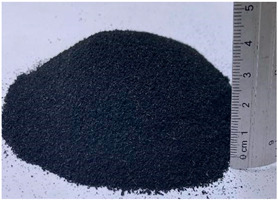
Sugar mill waste	Waste from sugarcane milling	Obtained from the sugar mill Casasano located in the municipality of Cuautla, Morelos, Mexico.	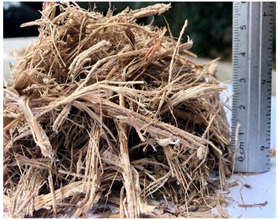

**Table 2 materials-15-06836-t002:** Procedures for the physical and chemical characterization of CDW.

Procedure	Physical Characteristics	Chemical Characteristics
[37]—Test method for particle size analysis in soils.	Granulometry	-----
[38]—Test method for the specific gravity of soils.	Density	-----
[39]—Tests to determine the chemical properties of aggregates (part 7).	-----	Chloride content soluble in water by Volhard’s method.
[39]—Tests to determine the chemical properties of aggregates (part 10).	-----	Sulfate content soluble in water.
[39]—Tests to determine the chemical properties of aggregates (part 11).	-----	Total sulfur content.

**Table 3 materials-15-06836-t003:** Partition Block base dosage.

Material	(% By Weight)
Construction waste “all in one”	80
Cement	5
Lime	10
Waste	2–5

**Table 4 materials-15-06836-t004:** Dosage of mixtures for the manufacture of the Partition Blocks.

Mixture Code	CDW % By Weight	PET % By Weight	Tires % By Weight	Bagasse % By Weight
M1	80.0	2.0	1.0	2.0
M2	78.0	3.0	1.0	3.0
M3	85.0	-	-	-
M4	83.0	-	2.0	-
M5	82.5	-	2.5	-
M6	82.0	-	3.0	-
M7	82.0	-	-	3.0
M8	81.0	-	-	4.0
M9	80.0	-	-	5.0
M10	80.0	5.0	-	-
M11	75.0	10.0	-	-
M12	70	15.0	-	-

**Table 5 materials-15-06836-t005:** Applicable standards for tests on bricks.

Specification	Mexican Norm
Compressive strength of blocks, bricks and pavers	[47]—NMX-036-ONNCCE-2013
Total water absorption and initial absorption	[48]—NMX-037-ONNCCE-2013
Size determination	[49]—NMX-038-ONNCCE-2013
Structural pieces	[51]—NMX-C-404-ONNCCE-2012
Non-structural pieces	[52]—NMX-C-441-ONNCCE-2013

**Table 6 materials-15-06836-t006:** Erodibility indices [53].

Parameter: Depth of Erosion D (mm/h)	
Rate erosion	Criteria
1	0 ≤ D ≤ 20
2	20 ≤ D ≤ 50
3	50 ≤ D ≤ 90
4	90 ≤ D ≤ 120

**Table 7 materials-15-06836-t007:** Results of physical and chemical characteristics of CDW (%).

Physical Characteristics (%)	
Granulometry	Sand = 94%, Fines = 6%
Density	2.9
Chemical Characteristics	
Content of soluble salts in soil	0.0010% < 0.01%
Water-soluble sulfate content [39]	0.0351% < 1%
Total sulfur content [39]	0.3285% < 1%

**Table 8 materials-15-06836-t008:** Results of physical and chemical characteristics of PET.

Physical Characteristics	Units	Values
Specific heat	(Kcal/Kg/°C)	0.25
Melting temperature	(°C)	25.5
Thermal conduction coefficient	(Kcal/m.h./°C)	0.25
Chemical Characteristics		Observation
Hydrocarbon resistance	----	Good 0.25
Effect of sun rays	----	Low affectation
Combustion behavior	----	Burns with medium difficulty

**Table 9 materials-15-06836-t009:** Composition of the sugarcane bagasse [58].

Parameter	Magnitude
**Chemical characteristics (%)**	
C	44.62
H	5.99
N	0.51
S	0.25
O	48.62
Cellulose	40.88
Hemicellulose	20.70
Lignin	22.80
Volatile matter	81.66
Fixed carbon	8.82
Ash	8.00, 5.10, 2.94
**Physical characteristics (%)**	
Density	184.48
Calorific power	7639.18
Humidity	4.42

**Table 10 materials-15-06836-t010:** Optimum mixture dosage for Partition Blocks (in weight percentage of solid material plus 1.4 L of water).

Mixture	CDW	PET	Cement	Lime	Water
	(Weight %)				(L)
M10-PET 5%	80	5	5	10	1.4

**Table 11 materials-15-06836-t011:** Dosage of the reference Partition Block (in weight percentage of solid material).

Mixture	CDW	Cement	Lime	Water
	(Weight %)			(L)
M3-CDW	85	5	10	1.4

**Table 12 materials-15-06836-t012:** Environmental impacts of conventional Partition Blocks and Partition Blocks made with CDW-PET 5%.

Conventional Partition Blocks	Partition Blocks Made with CDW-PET 5%
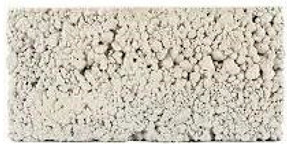	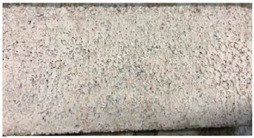
Virgin aggregates that increase the exploitation of material banks.	Recycled CDW aggregates available in large quantities and with low recycling or reuse levels.
Raw materials extraction accelerates their depletion or their extraction from more distant or less accessible sites.	Recycling of PET, which is one of the most abundant wastes.
Increased cost if aggregates are from more distant or less accessible sites.	Avoid the exploitation of natural banks.
Visual pollution when quarrying for aggregates.	Add CDW and PET to the value chains, preventing them from being immobilized in clandestine sites or landfills.
Increase in greenhouse gases.	Increase the life of landfills and final disposal sites.

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
