# Peer review of "Recovery of Mixtures of Construction Waste, PET and Sugarcane Bagasse for the Manufacture of Partition Blocks"

_materials, 2022, doi:10.3390/ma15196836_

Round 1
Reviewer 1 Report
In this article, the authors present the results of the feasibility study for the production of partition blocks (an essential building element) by combining construction and demolition waste (CDW), plastic flakes from polyethylene terephthalate (PET), dust from tire grinding and residues from the sugar industry (bagasse). The data obtained is interesting.
The publication requires a minor correction. Detailed comments:
1. There are many interesting ideas in the article, but the authors could better emphasize the novelty of the research carried out.
2. It is important to check that the writing text clearly expresses and explains each idea and result obtained.
3. Line 94 why is the font bold? Please remove the bold.
4. Line 400 and 408, please remove the bold.
Reviewer 2 Report
Reviewer’s Comments
The manuscript studied the recovery of mixtures of construction waste, pet and sugarcane bagasse for the manufacture of partition blocks. The topic is interesting and worth studying, and the reviewer suggests reconsidering the article after following revisions.
1, the ‘Introduction’ lacks the reviewing and summarizing of previous similar studies in construction and agricultural wastes. In the ‘introduction’, authors reported several key objective facts of three typical wastes. However, in my perspective, the picture is not fully portrayed. A myriad of researchers once attempted using various methods in modifying and recycling those wastes. The reviewer thereby suggests authors add those techniques in the article to demonstrate a more comprehensive portrait of waste disposal. The following articles might be helpful:
10.3390/ma14164417;
https://doi.org/10.1016/j.jobe.2022.104363
https://doi.org/10.1016/j.conbuildmat.2021.123901
2, please identify the novelty of this research.
3, some researchers once reported that the using of bio-wastes in concrete might cause impacts on the durability of end products. Were these issues taken into consideration in the study by authors? If not, please explain.
4, the using of ‘lime’ potentially triggers the high hydration heat and an early expansion of final products. How authors dealt with these issues?
5, in the ‘materials and methods’, please specify the mix. proportion of all samples. Besides, the curing conditions and time and other essential details of sample preparation processes should be provided.
6, basic physical properties and chemical compositions of raw materials should be provided.
7, in the introduction (line 86), authors classifies the constructions into ‘house’, ‘buildings’, etc.. Could authors explain it?
8, there are some typo errors in line 146. Please carefully check the entire manuscript and removal such errors throughout the entire paper.
9, what is the benchmark of the selection of proportion of RAs in each sample. Besides, the specific water content of each mixture is not reported.
10, it is better to replace the“absorption” by the ‘water absorption’. Plus, please reconsider the unit of the water absorption. What is the real meaning of the water absorption, is it the water absorption rate, water penetration rate, or the percentage of water absorbed by the sample? In addition, where is the 3.2.2?
11, the conclusion is poor. In the academic paper, the conclusion should contain the main findings of the results and the associated explanations of those findings. Please revise it.
Reviewer 3 Report
The article focuses on the feasibility of the manufacture of partition blocks (essential building element) through the combination of construction and demolition waste (CDW), polyethylene terephthalate (PET) plastic flakes, dust from tire shredding, and residue from the sugar industry (bagasse). Even though it seems the authors put in a lot of effort, the procedures and discussion of the experimental results did not meet the expectations of a scientific paper. In addition, compressive strength and absorption tests are not enough to evaluate the quality of construction material. Further tests (e.g., durability, mechanical properties) should be conducted. For all these reasons, unfortunately, I need to reject it. Specific comments can be found as follow:
· Introduction:
o The introduction cannot start with the focus of the paper. The introduction is meant to be from general to specific. Please modify accordingly.
o Lines 90-92: The gap in the paper is not clear. The authors should state what is done and what is not in the literature as well as the novelty of the publication.
· Materials and methods:
o Lines 102-103: I would call the stages in the same way as the subsections. Otherwise, it would hard for the reader to follow.
o Lines 127-131: I think that there is something wrong with this part. PET flakes or bagasse were not used as RA.
o Lines 158-161: the authors should list the water or water-to-cementitious materials used in the study. This is a very important factor.
o Lines 177-185: Why manual mixing? I think that manual mixing is much less consistent than mechanical mixing, especially with different operators. I am unsure about the manual mixing for a scientific paper.
o Line 260: Double-check this.
· Experimental results:
o Lines 329-341: I do not understand the differences between Figure 8 and Figure 10. Why, instead of adding a new figure, did you not include the standard deviations in Figure 8? Also, the subsection title 3.1.1 is the same as 3.1.
o Lines 391-414: This should be included in the Materials section.
· Discussion:
o The authors did not discuss the results of the experimental part. It looks more like a literature review instead of a discussion section.
· Conclusions and recommendations:
o Since there is no discussion, there is no good conclusions or recommendations. The authors basically did an overview of the results.
o Also, there is no mention of the economical cost until these sections. The calculation used should be included in the manuscript.
Round 2
Reviewer 2 Report
The authors assessed most of my comments, and greatly improved the quality of the article. the reviewer suggests accepting this article for publication after some minor considerations:
In the article, I still think the author did not explain clearly what measures were taken to avoid the effects of CaO after it was added, although it was explained in the reply.
In line 230,the ‘AR’ should be ‘RA’.
in line 325 and 326, the typography of the legend and text is out of order.
Reviewer 3 Report
The authors assessed most of my comments. The authors have greatly improved the quality of the article. I would recommend this article for publication after some minor considerations:
· Lines 266-275: I still believe that manual mixing is not the way to do it. It adds much more uncertainty to the results. Most of the laboratory experiments in concrete research are made using mechanical mixing even though at a small scale. You should consider changing this for future investigations.
· Lines 350-355: The compressive strength and the cross-sectional area cannot be both “A”. Change this.
